# Psychometric properties of the Chinese version of the Perinatal Bereavement Care Confidence Scale (C-PBCCS) in nursing practice

Jialu Qian[1,2], Honghe Wu[3], Shiwen Sun[2], Man Wang[1,2], Xiaoyan Yu[2]*

**1** Zhejiang University School of Medicine, Hangzhou, Zhejiang, China, **2** Department of Obstetrics, Women's Hospital School of Medicine, Zhejiang University, Hangzhou, Zhejiang, China, **3** Department of Obstetrics, Nantong Maternal and Child Health Hospital, Nantong, Jiangsu, China

* yuxy@zju.edu.cn

## Abstract

### Background

The Perinatal Bereavement Care Confidence Scale (PBCCS) was designed to evaluate midwives' and nurses' confidence and its psychosocial factors to provide bereavement care in Ireland. However, it is unknown whether this scale is valid and reliable for use with midwives and nurses in China. The aim of this study was to translate the English version into Chinese (C-PBCCS) and determine its validity and reliability in a population of Chinese midwives and nurses.

### Methods

In this cross-sectional observational study, after translating the English version of the PBCCS into Chinese and ensuring the linguistic adequacy and clarity of the language, we evaluated the validity and reliability of the C-PBCCS with Chinese midwives and nurses (n = 608). Participants were recruited using convenience sampling from 10 maternity hospitals in Zhejiang and Jiangsu Provinces. Exploratory factor analysis (EFA) was conducted to determine the construct validity (n = 304). Another sample of 304 midwives and nurses was used for confirmatory factor analysis (CFA) to verify the quality of the factor structures. Cronbach's alpha coefficient and Guttman split-half coefficient were adopted for the evaluation of internal consistency. The STROBE was followed in reporting the results.

### Results

The 43-item PBCCS was reduced to 40 items. Bereavement support knowledge (13 items, three factors), Bereavement support skills (eight items, two factors), Self-awareness (eight items, two factors), and Organizational support (11 items, two factors). The CFA suggested that the four scales in the C-PBCCS had acceptable fit indices. The Cronbach's alpha ranged from 0.835–0.901. The Guttman split-half coefficient was between 0.868–0.933.

**Data Availability Statement:** All relevant data are within the manuscript and its Supporting Information files.

**Funding:** The study was funded by Zhejiang Medical and Health Research Project (Foundation Number: 2020KY173). The funders had no role in study design, data collection and analysis, decision to publish, or preparation of the manuscript.

**Competing interests:** The authors have declared that no competing interests exist.

## Conclusion

The C-PBCCS was found to be a psychometrically sound measurement tool to evaluate Chinese-speaking midwives' and nurses' confidence and the psychosocial factors that affect their confidence in providing perinatal bereavement care.

## Introduction

Perinatal bereavement refers to the significant loss experiences of parents after perinatal loss [1]. Perinatal loss mainly included pregnancy loss and perinatal death. More specifically, perinatal loss encompasses miscarriage, therapeutic abortion (e.g., pregnancy termination due to foetal anomaly), stillbirth, or neonatal death [2]. The World Health Organization estimated that 43.8 million cases of pregnancies end in miscarriage [3]. The worldwide incidence of foetal anomalies is approximately 3.8% [4]. There are approximately 2.64 million cases of stillbirth, and 3.0 million cases of neonatal deaths occur globally [5]. Perinatal bereavement is a global health care issue. It is generally considered to be the most emotionally painful situation of bereavement because it is often sudden, unexpected, and in many cases unexplainable [6], which may cause serious and long-term psychological problems to bereaved parents [7].

The bereaved parents' experiences of clinical bereavement care may directly affect their ability to cope with perinatal loss and have a significant impact on their mental health [8]. Together, midwives and nurses form the largest group of professionals employed in health care services and potentially play an important role in bereavement care worldwide [9,10]. When the pregnancy ends with negative pregnancy outcomes, nursing professionals have the responsibility to help bereaved parents deal with this cruel reality; however, the majority of midwives and nurses are not prepared for this incident and lack the confidence to provide bereavement care [11–13]. Specifically, they reported inadequate counselling and communication skills [14]. They do not know how to use correct expression to comfort the women and communicate with their family [15]. The majority of health care professionals do not have a good command of professional knowledge in this special area [16,17]. In addition, insufficient support from colleagues and managers was reported [18,19]. Nursing staff pointed out that they felt lonely in providing perinatal bereavement care and suffered stigma from other colleagues and managers [20]. Thus, midwives and nurses are confronted with challenges, including inadequate knowledge and skills, as well as insufficient organizational support in perinatal bereavement care, which leads to their low confidence level [8,21,22].

The lack of perinatal bereavement care confidence among nursing professionals may lead to more obvious conflicts in clinical practice, such as lower service satisfaction and more serious emotional trauma of parents [23–25]. Therefore, it is important to focus on understanding the current status and influencing factors of midwives' and nurses' perinatal bereavement care confidence. Some studies and guidelines [26–32] have addressed the significance of providing continuous education and training to health care professionals to improve their confidence and ability to provide perinatal bereavement care. Previous studies mostly used self-designed questionnaires as evaluation tools to assess the effectiveness of education programmes [26]. It is necessary to provide a reliable assessment tool to measure improvement when midwives and nurses are participating in education and training in perinatal bereavement care. The perinatal bereavement care confidence scale (PBCCS) has been developed to evaluate midwives' and nurses' confidence and the psychosocial factors that affect their confidence in providing bereavement care [33]. It has been tested and applied in several Irish studies [21,34]. However,

there is currently a lack of effective tools for assessing the bereavement care confidence of nursing professionals in China. This problem urgently needs to be addressed to inform the education and training sector.

The PBCCS, a self-reported scale, was developed by Dr. Kalu [33]. It has been used with 277 midwives and nurses and is considered valid and reliable. However, an essential question is whether this scale can be used in other cultural contexts, such as for midwives and nurses in China. Therefore, this study was designed to determine the psychometric properties of the PBCCS among Chinese midwives and nurses and had two objectives: 1) to translate the English version of the PBCCS into Chinese and make cultural modifications and 2) to evaluate the validity and reliability of the Chinese version (C-PBCCS) for use with midwives and nurses from China.

## Methods

### Study design and participants

Participants were recruited from 10 hospitals with midwives and nurses. Cross-sectional, descriptive surveys were conducted in Zhejiang and Jiangsu Provinces, China. A convenience sampling was used. The inclusion criteria were participants who (a) were midwives and nurses working in the maternity ward or delivery room; (b) had working experience in providing perinatal bereavement care; and (c) were able to give their informed consent. The exclusion criterion was to be a nursing/midwifery student or have an intern in hospitals.

Data were collected between March and June 2021. This study was approved by the Ethics Committee of the Women's Hospital School of Medicine, Zhejiang University (IRB no. 20210091). All participants were informed of the aim of the study and guaranteed the confidentiality of private information and their right to withdraw from the study without any consequence.

### Instruments

**Demographic characteristic.** A self-designed form consisted of 10 questions about the participants' sociodemographic information, including age, current area of practice, education level, job title, marital status, having children, region, length of work experience, and training in perinatal bereavement care. This result indicated that the participants recruited are a representative sample of different backgrounds.

**The PBCCS.** The 43-item PBCCS developed by Felicity Agwu Kalu [33] measures midwives' and nurses' confidence and the psychosocial factors that influence their confidence in the provision of bereavement care to parents who experienced a perinatal loss. The original PBCCS consists of four scales: Scale A: perinatal bereavement support knowledge (15 items; Cronbach's alpha 0.833), Scale B: perinatal bereavement support skills (9 items; Cronbach's alpha 0.855), Scale C: self-awareness (8 items; Cronbach's alpha 0.797) and Scale D: organizational support (11 items; Cronbach's alpha 0.842). The scales of bereavement care knowledge and skills were used to assess midwives' and nurses' confidence. Self-awareness and organizational support were identified as psychosocial factors that influence confidence in providing bereavement care. The PBCCS uses a 5-point Likert scale ranging from 1 (strongly disagree) to 5 (strongly agree). Higher scores indicate higher confidence in the provision of bereavement care. The PBCCS has sound psychometric properties for use in Ireland [33].

**Translation and adaptation procedures and psychometric testing.** The English version of the PBCCS was translated into Chinese based on Brislin's translation model and the guidelines for the process of cross-cultural adaptation [35,36]. Permission to translate the scale was obtained from Dr. Kalu, the developer of the original PBCCS. Phase I involved three steps: 1)

Forward translation: two bilingual researchers independently translated the original PBCCS into Chinese. One was a nursing doctoral student familiar with the context on the scale; the other was a doctoral student in English and did not have a medical background. Dr. Kalu was consulted if there were any questions in the translation process. Any discrepancy in the two copies was reviewed and discussed with research team members, and a consensus was reached on a synthesis version. 2) Back-translation: The synthesis version was then back-translated by two experts who were completely blinded to the original PBCCS and were not working in the nursing field. Then, the two back-translations were compared and verified by the research team, from which a final Chinese translation was obtained. 3) Evaluation of content validity: A total of five experts were consulted to evaluate each item on a four-point Likert scale to confirm the content validity of each item and to determine whether these items were designed appropriately to measure the constructs. The expert panel included one specialist in maternity care (age, 57 years; working years, 40 years; job title, professor of nursing), two nurses in the maternity ward (one: age, 38 years; working years, 14 years; job title, nurse-in-charge; the other: age, 37 years; working years, 13 years; job title, nurse-in-charge), one midwife working in the delivery room (age, 34 years; working years, 10 years; job title, nurse-in-charge), and one professor in perinatal bereavement care (age, 52 years; working years, 35 years; job title, professor of nursing). Ambiguous terms were either removed or revised until no adjustments to the Chinese translation were considered necessary. We made some cultural modifications to the expression of certain items based on the experts' advice after two rounds of consultation. A content validity of an item (CVI) score above 0.8 was considered valid [37]. Some experts suggested deleting religion-related Item a9 from the bereavement support knowledge scale because most people in China do not believe in a religion. In regard to Item b8 from the bereavement support skills scale, some experts thought it was inappropriate for Chinese culture, as midwives and nurses rarely have the opportunity to come into contact with bereaved siblings. However, given that the original author believed that these items were important for perinatal bereavement care, we retained them for further analysis.

Phase II included two steps. 1) The revised version of the PBCCS was pilot tested to assess whether the C-PBCCS was easy to understand and answer with a convenience sample of 10 midwives and 10 nurses. 2) The psychometric properties of the C-PBCCS were estimated using item analysis, constructive validity, internal consistency reliability and split-half reliability. A total of 608 data points were collected and separated into two samples via a computerized random method adopting Excel. Sample 1, including the information from 304 midwives and nurses, was used for exploratory factor analysis (EFA). The other sample 2, including the information from 304 midwives and nurses, was used for confirmatory factor analysis (CFA) to verify the quality of the component structure. The internal consistency and stability of the scale were measured based on Cronbach's alpha coefficient and the Guttman split-half coefficient, respectively.

## Data collection

The head nurses of selected departments were approached through formal written letters that informed them about the purpose and procedures of the study. They issued questionnaires to midwives and nurses who met the eligibility requirements. The head nurses were responsible for explaining the matters needing attention during the distribution process. Data were collected using WJX (www.wjx.cn), a website that allows the creation of electronic questionnaires. A link to the survey was automatically generated and then sent to eligible midwives and nurses who consent to participate by WeChat (a chatting software). The questionnaire took approximately 20 minutes to complete. All participants gave their informed consent and were

guaranteed the confidentiality of private information. The online survey helped ensure that the submitted responses did not contain missing data.

## Statistical analysis

SPSS 21.0 IBM was used to conduct the data analysis. Demographic characteristics were presented using descriptive statistics, including the mean±standard deviation (SD) for continuous variables and frequencies and percentages for categorical variables. The internal consistency and homogeneity of the C-PBCCS were assessed using Cronbach's alpha (considering values ≥0.70 as appropriate) [38]. Item analysis was performed using the following analyses: (a) item-total correlation (b) extreme group comparison (the upper 27% and lower 27% scoring groups of an item should be able to discriminate) [39] (c) factor loading, and (d) Cronbach's alpha if an item was removed. Items that had a item-total correlation < 0.40, a criteria value (CR) < 3.0, factor loading < 0.40, and whose deletion led to an increase of 0.5 or more in the alpha coefficient for the overall scale were deleted.

The construct validity of the C-PBCCS was analysed by EFA. The Kaiser-Meyer-Olkin (KMO) test and Bartlett's spherical test were conducted prior to EFA (with values above 0.60 suggesting adequacy). The construct validity was evaluated by performing a principal component analysis with varimax rotation. The criterion for factor extraction was an eigenvalue > 1.0, and the inclusion of items was a factor loading > 0.40.

AMOS version 23.0 was employed to perform the CFA to further evaluate the validity of the C-PBCCS. A sample size of 200 was considered most appropriate for CFA [40]. Our sample size (n = 304) met the recommendation requirement for CFA. A standardized estimate should be greater than 0.45. A nonsignificant chi-square Index ($\chi^2$) was desirable. The following standards were used to evaluate model fit: $\chi^2$/degrees of freedom ratio (CMIN/DF) < 5.0 was deemed to be acceptable as marginal fit [41], root mean square residual (RMR) < 0.1, root mean square error of approximation (RMSEA) < 0.1, goodness-of-fit index (GFI) > 0.90, and adjusted goodness-of-fit index (AGFI) > 0.90 [42]. However, when in a large sample size, $\chi^2$ is often significant, and RMSEA is more suitable to assess the goodness of fit [43]. In addition, modification indices (MIs) were adopted to improve the fit of the model. To be similar to previous validation studies of PBCCS [21,33], four scales in the PBCCS were analysed.

## Results

### Sample characteristics

A convenience sample of 750 midwives and nurses completed the survey through WeChat; of them, 142 were excluded due to obviously irregular answers. Finally, 608 valid questionnaires were recovered, resulting in a response rate of 81%. The average age was 32.51 years (SD, 7.2). The majority of participants were nurses working in maternity wards (76.3%), undergraduates (83.2%), married (67.4%) and had not received training in perinatal bereavement care (85.2%). Detailed sample characteristics are displayed in Table 1.

### Psychometric analyses

**Face and content validity.** To evaluate face validity, the C-PBCCS was given to 10 midwives and 10 nurses to understand how they perceived and interpreted the items. The participants stated that the wording of the C-PBCCS was clear and that they had no difficulty understanding the scale. A general consensus was reached among the five experts that all the items in the questionnaire were relevant and comprehensive. The content validity index was 1.0. It indicates good validity [44].

**Table 1. Socio-demographic information of the participants (N = 608).**

| | Total sample | Sample 1 (N = 304) | Sample 2 (N = 304) |
|---|---|---|---|
| | n (%) or mean ± SD | n (%) or mean ± SD | n (%) or mean ± SD |
| Age (years) | 32.51±7.2 | 32.55±6.9 | 32.47±7.5 |
| Current area of practice | | | |
| Labour and birth | 144 (23.7) | 74 (24.3) | 70 (23.0) |
| Maternity ward | 464 (76.3) | 230 (75.7) | 234 (77.0) |
| Education level | | | |
| Secondary school | 4 (0.7) | 3 (1.0) | 1 (0.3) |
| Junior college | 93 (15.3) | 46 (15.1) | 47 (15.5) |
| Undergraduate | 506 (83.2) | 251 (82.6) | 255 (83.9) |
| Postgraduate or above | 5 (0.8) | 4 (1.3) | 1 (0.3) |
| Job title | | | |
| Primary nurse | 109 (17.9) | 52 (17.1) | 57 (18.8) |
| Nurse practitioner | 246 (40.5) | 121 (39.8) | 125 (41.1) |
| Nurse-in-charge | 198 (32.6) | 112 (36.8) | 86 (28.3) |
| Deputy director nurse | 55 (9.0) | 19 (6.3) | 36 (18.1) |
| Marital status | | | |
| Married | 410 (67.4) | 213 (70.1) | 197 (64.8) |
| Unmarried | 189 (31.1) | 89 (29.3) | 100 (32.9) |
| Divorced or widowed | 9 (1.5) | 2 (0.7) | 7 (2.3) |
| Having children | | | |
| Yes | 381 (62.7) | 192 (63.2) | 189 (62.2) |
| No | 227 (37.3) | 112 (36.8) | 115 (37.8) |
| Do you believe in a particular religion? | | | |
| Yes | 28 (4.6) | 13 (4.3) | 15 (4.9) |
| No | 580 (95.4) | 291 (95.7) | 289 (95.1) |
| Length of work experience | | | |
| < 2 years | 55 (9.0) | 26 (8.6) | 29 (9.5) |
| 2~5 years | 112 (18.4) | 58 (19.1) | 54 (17.8) |
| 5~10 years | 193 (31.7) | 94 (30.9) | 99 (32.6) |
| >10 years | 248 (40.8) | 126 (41.4) | 122 (40.1) |
| Training in perinatal bereavement care | | | |
| Yes | 90 (14.8) | 250 (82.2) | 36 (11.8) |
| No | 518 (85.2) | 54 (17.8) | 268 (88.2) |

## Item analysis

During the homogeneity analyses, the item-total correlations of Items a1, a14 and b5 were lower than 0.40 and did not meet the item retention criteria. After deleting these three items, the item-total correlation of each item ranged from 0.417 to 0.808 (see S1 File), suggesting moderate to strong correlations. The CR value was calculated to determine each item's degree of discrimination (values above 3.0 were desirable). All CR values were greater than 3.0 in our study (see S2 File).

## Construct validity and model fit

**Factor analysis of bereavement support knowledge scale.** To explore the best factor structure, EFA was utilized. A varimax rotation showed three factors with eigenvalues greater than 1.0 according to the KMO criteria. Bartlett's sphericity test was also conducted. The value

of the KMO test was 0.829, and Bartlett's test of sphericity was significant ($p < 0.001$), which indicated that the 13-item bereavement support knowledge scale was adequate for EFA. The construct validity of this scale revealed three distinct factors (confirmed by a scree plot; see S1 Fig in S3 File), with eigenvalues greater than 1.0 (4.448, 2.149 and 1.273). The factors found in the factor analysis accounted for 60.54% of the variance, with factor loadings varying from 0.561 to 0.854, all greater than 0.40, on all items (See S1 Table in S4 File). Therefore, the bereavement support knowledge scale was a three-factor structure in the context of Chinese culture. Factor 1 included six items related to "general knowledge of bereavement support needs": Items a4, a5, a9, a10, a12, and a13. Factor 2 contained three items related to "knowledge limitation": Items a7, a8, and a11. Factor 3 included four items related to "professional meaning and knowledge development": Items a2, a3, a6, and a15. A structure matrix showed that the factor loading of Item a12 was greater than 0.40 in both Factor 1 and Factor 2. However, Item a12 was included in Factor 1 because its content was highly correlated with general knowledge of bereavement support needs.

The three-factor model was also chosen to conduct CFA using another dataset from a sample of 304 participants. The CFA results demonstrated that all 13 items had met the requirement of a standardized estimate ($\geq 0.45$). The present model containing 13 items and three factors provides an acceptable fit to the data: $\chi^2/df = 3.988$, RMR = 0.065, RMSEA = 0.099, GFI = 0.880, AGFI = 0.824 (See Fig 1).

**Factor analysis of bereavement support skills scale.** The KMO measure of the bereavement support skills scale was 0.845, which was better than the minimal admissible level of 0.60. Bartlett's test of sphericity was also significant ($p < 0.001$), which indicated that the 8-item bereavement support skills scale was suitable for EFA. The construct validity of this scale revealed two distinct factors (confirmed by a scree plot; see S2 Fig in S3 File), with eigenvalues greater than 1.0 (4.085 and 1.035). The factors found in the factor analysis accounted for 64.00% of the variance, with factor loadings varying from 0.657 to 0.829, all greater than 0.40, on all items (See S2 Table in S4 File). Therefore, in the context of Chinese culture, the bereavement support skills scale may have two factors. Factor 1 contained four items related to "bereavement support specific interpersonal skills": Items b6, b7, b8, and b9. Factor 2 included four items related to "bereavement functional skills": Items b1, b2, b3, and b4. A structure matrix showed that the factor loadings of Item b1 and b4 in both Factor 1 and Factor 2 were greater than 0.40. The factor loading values of Items b1 and b4 in Factor 2 were greater than the factor loading values in Factor 1, and their contents were more important for Factor 2 as their contents addressed bereavement functional skills.

All eight items had met the requirement of a standardized estimate ($\geq 0.45$). The result of CFA testing the two-factor structure had acceptable fit indices: $\chi^2/df = 4.692$, RMR = 0.036, RMSEA = 0.110, GFI = 0.933, AGFI = 0.874 (See Fig 2).

**Factor analysis of the self-awareness scale.** The KMO measure of the self-awareness scale was 0.832, and Bartlett's test of sphericity was admissible ($p < 0.001$). This result suggested that the self-awareness scale was adequate for EFA. Two distinct factors were revealed (confirmed by a scree plot; see S3 Fig in S3 File), with eigenvalues greater than 1.0 (3.927 and 1.112). The factors found in the factor analysis explained 62.98% of the variance, with factor loadings ranging from 0.514 to 0.855 (See S3 Table in S4 File). The self-awareness scale had eight items, reflecting two factors. Factor 1 included five items related to "competence and resource awareness": Items c1, c2, c3, c6, and c7, and Factor 2 contained three items related to "deficiency and learning awareness": Items c4, c5, and c8. A structure matrix showed that the factor loadings of Items c2 and c3 in both Factor 1 and Factor 2 were greater than 0.40. However, these two items were included in Factor 1 because their contents were closely associated with competence and resource awareness.

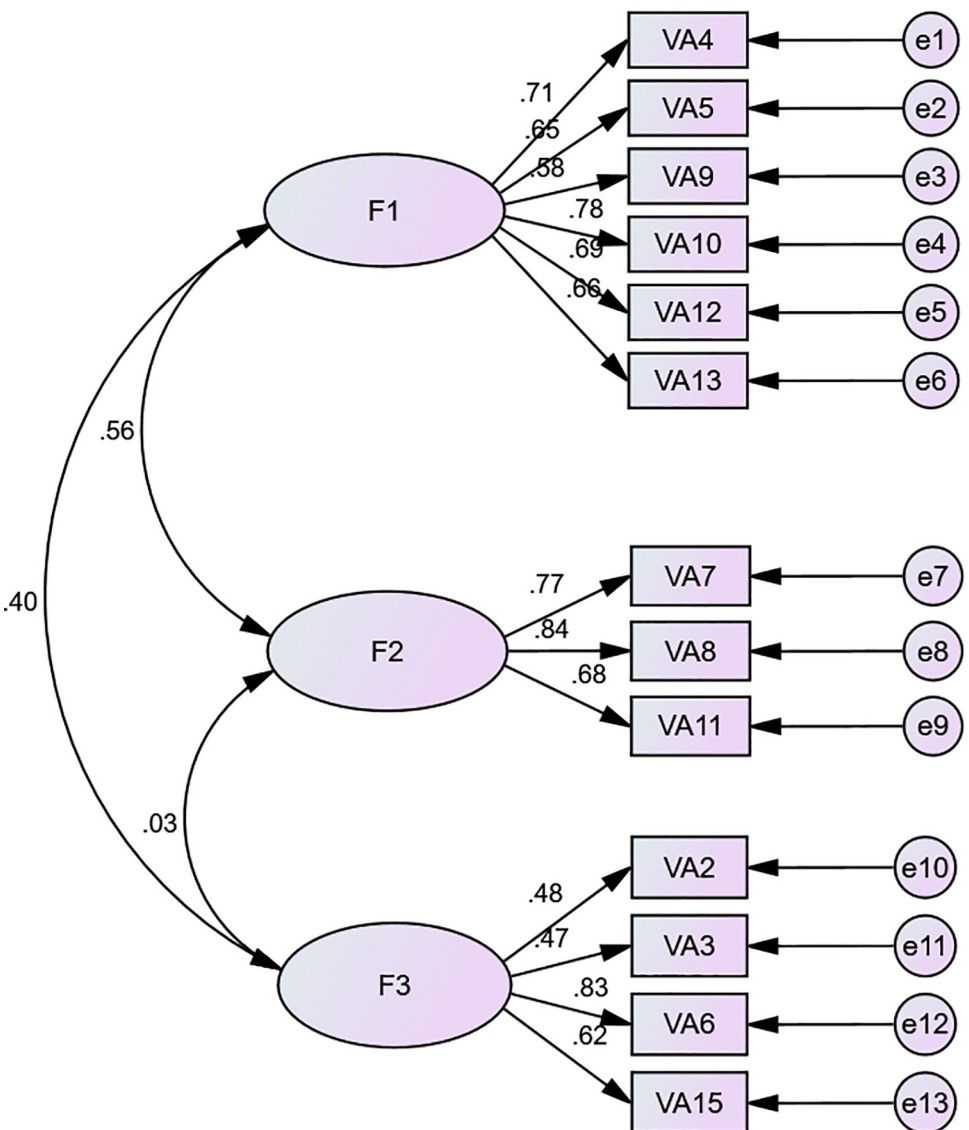

**Fig 1. Confirmatory factor analysis of the bereavement support knowledge scale.** F1: General knowledge of bereavement support needs; F2: Knowledge limitation; F3: Professional meaning and knowledge development.

The two-factor model was chosen to conduct CFA. The CFA results demonstrated that all eight items had met the requirement of a standardized estimate ($\geq 0.45$). Therefore, the present model including eight items and two factors provides a acceptable model to the data: $\chi^2$/df = 4.518, RMR = 0.034, RMSEA = 0.108, GFI = 0.934, AGFI = 0.875 (See Fig 3).

**Factor analysis of organizational support scale.** The value of the KMO test was 0.886, and Bartlett's test of sphericity was significant (p < 0.001), which indicated that the 11-item organizational support scale was adequate for EFA. The construct validity of this scale revealed two distinct factors (confirmed by a scree plot; see S4 Fig in S3 File), with eigenvalues greater than 1.0 (5.707 and 1.498). The two factors found in the factor analysis accounted for 65.50% of the variance, with factor loadings varying from 0.622 to 0.915 (See S4 Table in S4 File). Therefore, in the context of Chinese culture, the organizational support scale may have two factors. Factor 1 included nine items related to "support for staff providing bereavement care":

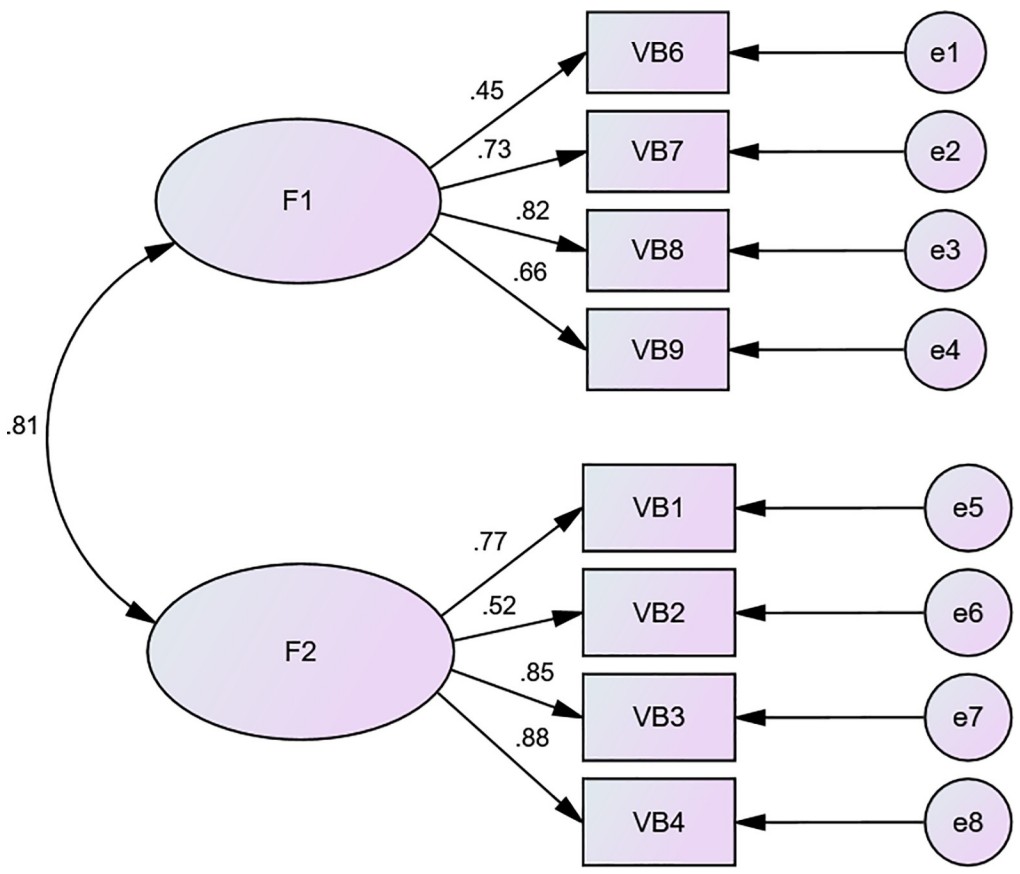

**Fig 2. Confirmatory factor analysis of the bereavement support skills scale.** F1: Bereavement support specific interpersonal skills; F2: Bereavement functional skills.

Items d1, d2, d3, d4, d5, d7, d8, d9 and d10, and Factor 2 contained two items related to "workload influences": d6 and d11.

The CFA was performed on the two-factor structure model. The initial model indices suggested a poor fit. [$\chi 2/df = 6.798$, RMR = 0.041, RMSEA = 0.138, GFI = 0.842, AGFI = 0.758]. The modification indices suggested estimation of the error covariances between Item d1 and Item d2. When we respecified the model to include this error covariance, it indicated an improvement in all the indices and achieved a better satisfactory fit: $\chi 2/df = 3.681$, RMR = 0.035, RMSEA = 0.094, GFI = 0.906, AGFI = 0.853 (See Fig 4).

## Internal consistency and split-half reliability

For the four scales of the C-PBCCS, including bereavement support knowledge and skills, self-awareness, and organizational support, the corresponding Cronbach's alpha coefficients were 0.835, 0.862, 0.852, and 0.901, respectively. S5 File shows the Cronbach's alphas if an item was deleted, means and SD. The respective Guttman split-half coefficients of the four scales were 0.878, 0.906, 0.868 and 0.933. The above results indicated adequate internal consistency and reliability.

## Discussion

Relevant guidelines and principles identify the importance of health care professionals' education in perinatal bereavement care provided for women and their families following perinatal

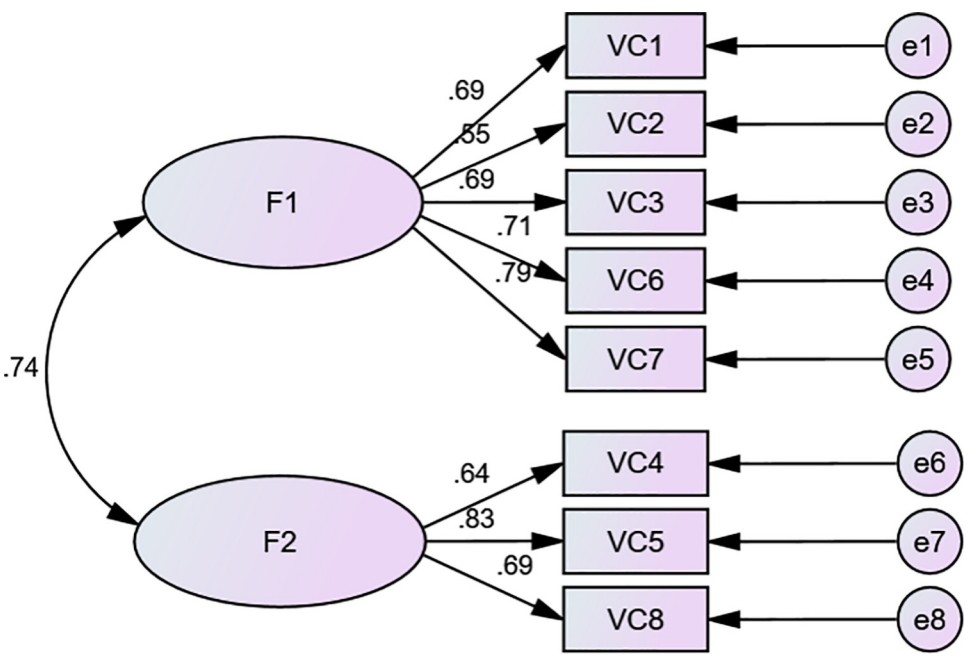

**Fig 3. Confirmatory factor analysis of the self-awareness scale.** F1: Competence and resource awareness; F2: Deficiency and learning awareness.

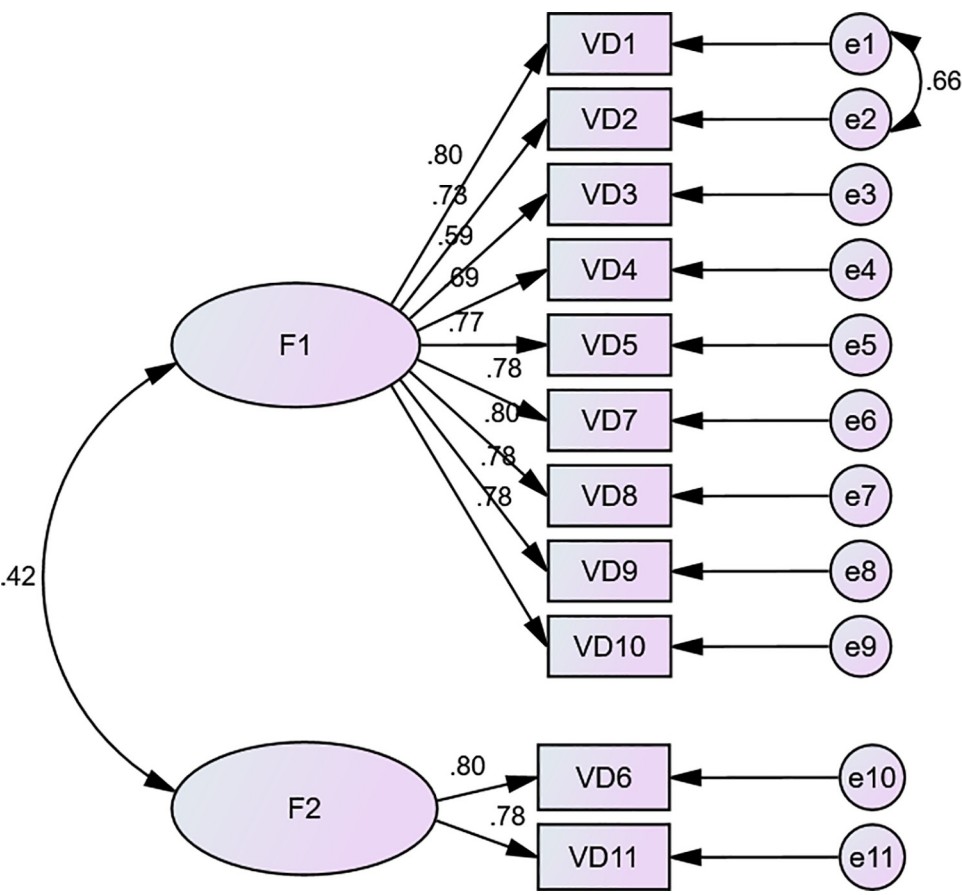

**Fig 4. Confirmatory factor analysis of the organizational support scale.** F1: Supporting degree; F2: Workload influences.

loss [27–29]. Midwives and nurses need to be equipped with confidence to provide perinatal bereavement care [21,45–48]. Therefore, it is essential to find a valid and practical measurement to measure midwives' and nurses' confidence and the psychosocial factors that impact the provision of bereavement care, especially in China. The aim of the present study was to translate the English version of the well-validated PBCCS into Chinese and to examine its reliability and validity.

The participants of this study were recruited from 10 different levels of maternity hospitals. The use of WeChat for data collection overcame the space limitation. It allowed us to include midwives and nurses of diverse backgrounds. Overall, the C-PBCCS showed good face validity, content validity, construct validity, internal consistency, and split-half reliability in our study, which demonstrated that the 40-Item C-PBCCS was a reliable measurement. Furthermore, another set of data from 304 participants was used to conduct a CFA. The results indicated that the response data fit well with the hypothetical structure of the four scales in the C-PBCCS, which provided positive evidence for its construct validity. We propose that the C-PBCCS is an appropriate tool for assessing the confidence of midwives and nurses in providing perinatal bereavement care in mainland China.

The translation and adaptation of C-PBCCS were undertaken strictly following the methodology guidelines [49]. Most items appeared to have culturally equivalent terms in Chinese. Hence, we could translate the PBCCS without extensive adaptation. We sought the advice of Dr. Kalu, the developer of PBCCS, on translation and cross-cultural adaptation. Two items were modified during the translation process. One was the definition of the "social needs of bereaved parents" in Item a6. To make the item easier to understand by nursing professionals, "I understand the social needs of bereaved parents" was modified to "I understand the reality of loss should be socially defined as significant". The other modification was that for cultural adaptation and better understanding, we specified the definition of "cultural needs of bereaved parents" in Item a5. We used "I understand the needs of bereaved parents from different cultural and ethnic backgrounds" in place of "I understand the cultural needs of bereaved parents". It is worth noting that some experts consider deleting religion-related Item a9 from the bereavement support knowledge scale because most people in mainland China are nonreligious [50]. However, the research team found that bereaved parents of different cultural backgrounds have different attitudes towards perinatal loss, and it is necessary to provide cultural care for bereaved parents based on their beliefs and needs [31,51]. Additionally, indeed, some Chinese believe in Buddhism and Christianity. The scale retained items related to cultural needs. Hence, the scale retained this item related to religion.

Compared with the original English version of the 43-item PBCCS, the items were reduced to 40 in the C-PBCCS, and it performed well on four scales. The structures of the four scales were different from those of the original Ireland scales. The C-PBCCS can be seen in S5 File. In regard to the bereavement support knowledge scale, we identified three factors through EFA, and three factors were retained via CFA: general knowledge of bereavement support needs (Items a4, a5, a9, a10, a12, and a13), knowledge limitations (Items a7, a8, and a11) and professional meaning and knowledge development (Items a2, a3, a6, and a15), which was different from the original scale. Two items, a1 and a14 were not satisfactory and did not meet the criteria of item retention. The bereavement support skills scale had two factors in EFA, and retained two factors structure in CFA. The two-factor model included eight items and two dimentions: bereavement support specific interpersonal skills (Items b6, b7, b8, and b9) and bereavement functional skills (Items b1, b2, b3 and b4), which was basically similar to the original scale. One item (b5) was removed in item analysis. The present 8-item scale could assess the practical skills of providing psychological support, information support, and satisfying the needs of bereaved parents expecting their next baby. These items mostly covered the assistance

that bereaved parents required from midwives and nurses, as proposed in previous studies [52–54]. The self-awareness scale contained 8 items and two factors: competence and resource awareness (Items c1, c2, c3, c6 and c7) and deficiency and learning awareness (Items c4, c5 and c8). The original scale addressed the "awareness of the needs of bereaved families" and "awareness of my personal needs in relation to providing bereavement support". In the context of Chinese culture, we named the two factors from the perspective of focusing on awareness related to midwives and nurses given that the understanding of bereaved families' needs has already been included in the bereavement support knowledge scale (general knowledge of bereavement support needs). Therefore, the results suggested that it was more reasonable to focus on the awareness of nursing professionals themselves. In regard to the organizational support scale, the scale structure was different from the original Ireland scale. It also included two factors and 11 items: support for staff providing bereavement care (Items d1, d2, d3, d4, d5, d7, d8, d9 and d10) and workload influences (d6 and d11). We specified the covariance between residuals of Items d1 and d2 to fit the data. Generally, the sufficient support received by nursing professionals from managers in providing perinatal bereavement support means that managers attached importance to this issue. Therefore, managers would demand that other providers pay attention to providing bereavement support, so providers could receive support from their colleagues. In these contexts, these covariances are plausible.

The reliability evaluation showed that the C-PBCCS was an acceptable instrument in midwives and nurses. The Cronbach's alpha coefficients of the four scales, with values of 0.835, 0.862, 0.852, and 0.901, showed good internal consistency; the correlations of 0.878, 0.906, 0.868 and 0.933 of the split-half internal consistency test also indicated the sound reliability of the C-PBCCS. The findings of this study were similar to the results for the original English version, in which Cronbach's alpha coefficients ranged from 0.797 to 0.855 in midwives and nurses in Ireland [33]. In addition, this study found that all 40 items had adequate discrimination. The psychometric characteristics of the C-PBCCS verified the stability and internal consistency of the measurement.

Our results revealed that Item a11 "I do not have adequate practical knowledge for bereavement support", Item b2 "I do not have adequate perinatal bereavement support experience", Item c1 "I am aware of the needs of recently bereaved parents" and Item d7 "There is a clear policy in my ward/unit for the provision of bereavement support to parents" had the lowest scores on the four scales. This result indicated insufficient emphasis on bereavement support, a lack of clear policies and deficiencies in professional knowledge and the experience of bereavement support, which were in line with previous findings [16,17,55].

## Limitations

The current study has several limitations. First, the sample of midwives and nurses was mainly from the Zhejiang and Jiangsu Provinces of China. It is limited to using a convenience sampling method; thus, the findings may not represent the opinions of all midwives and nurses in China. Second, since the C-PBCCS is a self-reported instrument, a social desirability bias may exist in the responses. Although the investigation ensured the anonymity of participants, midwives and nurses may have chosen responses that conformed to managers' expectations. This possibility is caused by the special cultural background in China. Department managers have authority on work arrangements and employee bonuses. Moreover, leaders usually hope that their organizations could have a positive image. Consequently, employees may subconsciously choose responses consistent with the expectations of their managers. Third, there was no other reported scale available in the literature to measure midwives' and nurses' confidence. We did not calculate the concurrent validity evaluation in this study.

### Implication for future research

To the best of our knowledge, this is the first study to evaluate the psychometric properties of the PBCCS in China. The factor structure examined in the study may be affected by the norm and social culture of the sample. Therefore, the present results need to be replicated in other cultural contexts for further validation to enhance the generalizability of this scale. The findings provide further support for the validity and reliability of the C-PBCCS for the measurement of midwives' and nurses' bereavement support knowledge, bereavement support skills, self-awareness, and organizational support. Future studies may attempt to establish a structural equation model (SEM) to analyse the potential influencing factors of perinatal bereavement confidence and their correlations. Future studies can also test whether this instrument is useful with multidisciplinary health care professionals (doctors and psychologists, for example).

### Implication for clinical practice

There is increasing acknowledgment of the importance and value of enhancing midwives' and nurses' perinatal bereavement confidence and ability in practice. Relevant studies and guidelines have addressed the significance of improving bereavement support services by strengthening support and continuous education to health care professionals [27,28,31,32]. The C-PBCCS may assist clinical nursing managers in exploring staff perceptions and influencing factors of providing perinatal bereavement support. Findings from the C-PBCCS could inform the inadequate capabilities of nursing professionals and limitations in clinical practice, which provide improvement directions for future practice and the development of training program content. Another potential use of the PBCCS would be as an assessment tool for education and training programmes related to perinatal bereavement care among midwives and nurses to evaluate the effectiveness of such education programmes in bringing about change in their confidence.

### Conclusions

The results of this study showed that the 40-Item C-PBCCS has satisfactory face, content, and construct validity, as well as internal consistency. The findings of the cross-sectional study confirmed that it was a helpful instrument to assess mainland Chinese midwives' and nurses' confidence regarding perinatal bereavement care and relevant psychosocial factors. Further evidence supporting its application is expected from a larger sample that is more representative of the Chinese nursing population or other health care professionals.

### Supporting information

**S1 File. The item-total correlations of all items.**
(DOC)

**S2 File. The criterial value (CR) of all items.**
(DOC)

**S3 File. Scree plot of the C-PBCCS.**
(DOC)

**S4 File. Rotated factor analysis of the C-PBCCS.**
(DOC)

**S5 File. The results of factor analysis of the four scales of C-PBCCS.**
(DOC)

**S6 File. STROBE statement.**
(DOC)

**S7 File. Minimal data set.**
(XLSX)

## Acknowledgments

The authors would like to acknowledge and thank all study participants.

## Author Contributions

**Conceptualization:** Jialu Qian.

**Data curation:** Honghe Wu.

**Formal analysis:** Shiwen Sun, Man Wang.

**Funding acquisition:** Xiaoyan Yu.

**Investigation:** Jialu Qian, Honghe Wu.

**Methodology:** Shiwen Sun, Man Wang.

**Software:** Man Wang.

**Supervision:** Xiaoyan Yu.

**Writing – original draft:** Jialu Qian.

**Writing – review & editing:** Jialu Qian, Honghe Wu, Shiwen Sun, Man Wang, Xiaoyan Yu.

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
