## [Decision Letter · Decision Letter 0]

30 Dec 2021

PONE-D-21-35623Psychometric properties of the Chinese version of the Perinatal Bereavement Care Confidence Scale (C-PBCCS) in nursing practicePLOS ONE

Dear Dr. Yu,

Thank you for submitting your manuscript to PLOS ONE. After careful consideration, we feel that it has merit but does not fully meet PLOS ONE’s publication criteria as it currently stands. Therefore, we invite you to submit a revised version of the manuscript that addresses the points raised during the review process.

ACADEMIC EDITOR:Thank you for inviting me to review this manuscript. The translation and validation of an instrument to assess the perinatal bereavement care confidence of nurses and midwives is important to develop additional education and support for them to provide bereavement care for the patients. I have the following comments and suggestions for revision:

1. In Line 49, the symbol "~" should be revised to "-".

2. In Line 141, please add the meaning of higher scores indicating higher or lower confidence in the provision of bereavement care.

3. In Line 142, please add the reference.

4. in Line 149, the word "a nursing doctor student" should be "a nursing doctoral student".

5. In Line 160, the word "create the constructs" should be "measure the constructs".

6. in Line 171, the word "region" should be "religion".

7. In Line 232, the title of Table 1 should be "Socio-demographic information ......"

8. In Line 247, what's the meaning of "CR value"? Please provide more information.

9. In Line 285, "See Additional file 4" should be "See S4 File".

10. In Line 307, "See Additional file 4" should be "See S4 File".

11. In Line 346, S5 File is referred. What's the meaning of highlighted 4 items in S5 File? And there are two S5 Files are uploaded. Is there any difference between the two files?

12. In Line 477, please add the wording assessment tool "for" education.

We look forward to receiving your revised manuscript.

Kind regards,

Ka Ming Chow

Academic Editor

PLOS ONE

Reviewers' comments:

Reviewer's Responses to Questions

**Comments to the Author**

1. Is the manuscript technically sound, and do the data support the conclusions?

Reviewer #1: Yes

2. Has the statistical analysis been performed appropriately and rigorously? 

Reviewer #1: Yes

3. Have the authors made all data underlying the findings in their manuscript fully available?

Reviewer #1: Yes

4. Is the manuscript presented in an intelligible fashion and written in standard English?

Reviewer #1: Yes

5. Review Comments to the Author

Reviewer #1: Line 64 Add emotionally before painful

Line 82; Low confidence is noted, but no citation to support this.

Line 149; change ‘nursing doctor student’ to doctoral nursing student

Line 160: Were the nursing experts excluded from the study?

Line 171 Update region to religion

Line 189-193: How was it identified who would be getting the survey from this website? The total participants who completed, is this the total number of individuals who are part of the WeChat? More clarification is needed in this area.

Line 226-227: A total of 750 midwives and nurses (is this the total number of WeChat participants?) Is WeChat only for maternity nurses?

Table 1 Education Level (can information be added to include years or equivalence to the United States educational pathways?)

Table 1 Personality~ Individuals self-selected introverted, neutral, and extroverted. Was a tool offered for individuals to complete to identify this? As self-selecting and completing of a tool may result in different outcomes.

Line 356 Nurses’ and midwives’ written, rather than the consistent order midwives and nurses throughout manuscript.

Line 360 Here you note recruited from 10 different areas. More information is needed to fully describe the connection between recruiting and WeChat use.

Line 400 update ‘items’ to ‘item’

This study is value added information for maternity healthcare providers. Families all over the world experience perinatal loss, and healthcare providers need the tools and confidence to provide care to these families. The translation and application of this tool to a variety of languages will enable educators and leaders to implement training and potentially simulation scenarios to further develop nurses’ ability to provide specific perinatal loss care to families.

6. PLOS authors have the option to publish the peer review history of their article (what does this mean?). If published, this will include your full peer review and any attached files.

Reviewer #1: **Yes: **Dr. Sabrina Ehmke DNP, RNC-OB, NPD-BC, PHN

---

## [Author Response · Author response to Decision Letter 0]

3 Jan 2022

Response to academic editor:

General comment:

Thank you for inviting me to review this manuscript. The translation and validation of an instrument to assess the perinatal bereavement care confidence of nurses and midwives is important to develop additional education and support for them to provide bereavement care for the patients.

Reply:

We appreciate the positive feedback from the editor. We would like to thank the editor for careful and thorough reading of this manuscript, which help to improve the quality of this manuscript. Our response follows (the reviewer’s comments are in italics).

Specific comments:

Comment 1:

In Line 49, the symbol "~" should be revised to "-".

Reply:

Thank you for the editor’s careful reviewing. The correction has been made as suggested. Please see line 49.

Comment 2:

In Line 141, please add the meaning of higher scores indicating higher or lower confidence in the provision of bereavement care.

Reply:

We have added the sentence “Higher scores indicate higher confidence in the provision of bereavement care ” in the revised manuscript. Please see line 142.

Comment 3:

In Line 142, please add the reference.

Reply:

As suggested, we have added the reference. Please see line 143.

Comment 4:

in Line 149, the word "a nursing doctor student" should be "a nursing doctoral student".

Reply:

The correction has been made.

Line 150: ‘a nursing doctor student’ -> ‘a nursing doctoral student’

Comment 5:

In Line 160, the word "create the constructs" should be "measure the constructs".

Reply:

The suggested correction has been made.

Line 161: ‘create the constructs’ -> ‘measure the constructs’

Comment 6:

in Line 171, the word "region" should be "religion".

Reply:

The suggested correction has been made.

Line 172: ‘region’-> ‘religion’

Comment 7:

In Line 232, the title of Table 1 should be "Socio-demographic information ......"

Reply:

The suggested correction has been made. Please see line 239.

Comment 8:

In Line 247, what's the meaning of "CR value"? Please provide more information.

Reply:

We have added a description of the meaning of CR value. Please see line 254. The text now reads:

‘CR value was calculated to determine each item’s degree of discrimination (values above 3.0 were desirable).’

Comment 9:

In Line 285, "See Additional file 4" should be "See S4 File".

Reply:

The suggested correction has been made.

Line 294: ‘See Additional file 4’-> ‘See S4 File’

Comment 10:

In Line 307, "See Additional file 4" should be "See S4 File".

Reply:

The suggested correction has been made.

Line 316: ‘See Additional file 4’-> ‘See S4 File’

Comment 11:

In Line 346, S5 File is referred. What's the meaning of highlighted 4 items in S5 File? And there are two S5 Files are uploaded. Is there any difference between the two files?

Reply:

Highlighted 4 items had the lowest scores on the four scales. We described these four items in the Discussion. Please see line 440-446.

We are sorry that we mistakenly uploaded two S5. We only have one S5 File in our study.

Comment 12:

In Line 477, please add the wording assessment tool "for" education.

Reply:

The suggested correction has been made. Please see line 486.

Response to Review #1:

Comment 1:

Line 64 Add emotionally before painful

Reply:

Thank you for the reviewer’s careful reviewing. We have added ‘emotionally’ before ‘painful’. Please see line 64. 

Comment 2:

Line 82; Low confidence is noted, but no citation to support this.

Reply:

As suggested, we have added three relevant references to support it. Please see line 83.

Comment 3:

Line 149; change ‘nursing doctor student’ to doctoral nursing student

Reply:

The correction has been made.

Line 150: ‘a nursing doctor student’ -> ‘a nursing doctoral student’

Comment 4:

Line 160: Were the nursing experts excluded from the study?

Reply:

We included nurses and midwives with different education level, job title and length of work experience. Therefore, nursing professionals with rich clinical experience as well as young nurses were all included in this study.

Comment 5:

Line 171 Update region to religion

Reply:

The suggested correction has been made.

Line 172: ‘region’-> ‘religion’

Comment 6:

Line 189-193: How was it identified who would be getting the survey from this website? The total participants who completed, is this the total number of individuals who are part of the WeChat? More clarification is needed in this area.

Reply:

Thank you for the reviewer’s valuable comment. The head nurses were responsible for distributing questionnaires to midwives and nurses who met the eligibility requirements. 

All the participants completed the questionnaire through WeChat. A total of 750 questionnaires were distributed in this study; of them, 142 were excluded due to obviously irregular answers. Finally, 608 valid questionnaires were recovered.

We have revised the section of “Data collection” to give more information. Please see line 189-199.

Comment 7:

Line 226-227: A total of 750 midwives and nurses (is this the total number of WeChat participants?) Is WeChat only for maternity nurses?

Reply:

Line 232-234: A convenience sample of 750 midwives and nurses completed the survey through WeChat; of them, 142 were excluded due to obviously irregular answers. Finally, 608 valid questionnaires were recovered, resulting in a response rate of 81%. 

WeChat was not only for maternity nurses. The link of the questionnaire were sent to midwives and nurses working in the maternity ward or delivery room using WeChat.

Comment 8:

Table 1 Education Level (can information be added to include years or equivalence to the United States educational pathways?)

Reply:

Thank you for the reviewer’s comment. Given that this survey was conducted in mainland China, we collected information of education level according to Chinese educational pathways. Therefore, we could not add information to include years or equivalence to the United States educational pathways.

Comment 9:

Table 1 Personality~ Individuals self-selected introverted, neutral, and extroverted. Was a tool offered for individuals to complete to identify this? As self-selecting and completing of a tool may result in different outcomes.

Reply:

Thank you for the reviewer’s helpful comment. In this study, we just wanted to have a general idea about midwives’ and nurses’ personality and it was not the research emphasis of our study. Therefore, we did not use a tool for personality identification. Participants chose the personality based on their subjective judgments. 

Comment 10:

Line 356 Nurses’ and midwives’ written, rather than the consistent order midwives and nurses throughout manuscript.

Reply:

The correction has been made as suggested.

Line 364: Nurses’ and midwives’-> midwives’ and nurses’

Comment 11:

Line 360 Here you note recruited from 10 different areas. More information is needed to fully describe the connection between recruiting and WeChat use.

Reply:

The advantage of WeChat use in the recruitment has been added. Please see line 369. The text now reads:

‘The use of WeChat for data collection made up the limitation of space so that we could include midwives and nurses of diverse backgrounds. ’

Comment 12:

Line 400 update ‘items’ to ‘item’

Reply:

The suggested correction has been made.

Line 409: ‘items’-> ‘item’

General comment:

This study is value added information for maternity healthcare providers. Families all over the world experience perinatal loss, and healthcare providers need the tools and confidence to provide care to these families. The translation and application of this tool to a variety of languages will enable educators and leaders to implement training and potentially simulation scenarios to further develop nurses’ ability to provide specific perinatal loss care to families.

Reply:

We appreciate the positive feedback from the reviewer. We would like to thank the reviewer for careful and thorough reading of this manuscript and for the thoughtful comments and constructive suggestions, which help to improve the quality of this manuscript.

---

## [Editor Report · Decision Letter 1]

5 Jan 2022

PONE-D-21-35623R1Psychometric properties of the Chinese version of the Perinatal Bereavement Care Confidence Scale (C-PBCCS) in nursing practicePLOS ONE

Dear Dr. Yu,

Thank you for submitting your manuscript to PLOS ONE. After careful consideration, we feel that it has merit but does not fully meet PLOS ONE’s publication criteria as it currently stands. Therefore, we invite you to submit a revised version of the manuscript that addresses the points raised during the review process.

ACADEMIC EDITOR:The authors have revised the manuscript and addressed most comments from the reviewers. Two minor comments are provided for further improve the quality of the manuscript: 1. In Table 1, personality of the participants was reported. How to categorise the participants into three different personalities? Which instrument was adopted for assessment? What's the significance of this data to the results?2. Some grammatical errors which affect the clarity, especially the revised and newly added parts.

We look forward to receiving your revised manuscript.

Kind regards,

Ka Ming Chow

Academic Editor

PLOS ONE
---

## [Author Response · Author response to Decision Letter 1]

6 Jan 2022

Response to academic editor

Specific comments:

Comment 1:

In Table 1, personality of the participants was reported. How to categorise the participants into three different personalities? Which instrument was adopted for assessment? What's the significance of this data to the results?

Reply:

Thank you for the editor’s valuable comment. We did not use an instrument to assess participants’ personality. Participants chose personality according to their subjective judgements. Given that participants’ personality has little relation with the results in our study, we have removed it from Table 1 to avoid ambiguity. Please see Table 1.

Comment 2:

Some grammatical errors which affect the clarity, especially the revised and newly added parts.

Reply:

Thank you for the editor’s careful reviewing. We have invited two native speakers to polish the revised manuscript. Some grammatical errors have been corrected.

---

## [Editor Report · Decision Letter 2]

10 Jan 2022

Psychometric properties of the Chinese version of the Perinatal Bereavement Care Confidence Scale (C-PBCCS) in nursing practice

PONE-D-21-35623R2

Dear Dr. Yu,

We’re pleased to inform you that your manuscript has been judged scientifically suitable for publication and will be formally accepted for publication once it meets all outstanding technical requirements.

Kind regards,

Ka Ming Chow

Academic Editor

PLOS ONE

---

## [Editor Report · Acceptance letter]

13 Jan 2022

PONE-D-21-35623R2 

Psychometric properties of the Chinese version of the Perinatal Bereavement Care Confidence Scale (C-PBCCS) in nursing practice 

Dear Dr. Yu:

I'm pleased to inform you that your manuscript has been deemed suitable for publication in PLOS ONE. Congratulations! Your manuscript is now with our production department. 

Kind regards, 

on behalf of

Dr. Ka Ming Chow 

Academic Editor

PLOS ONE